# Quantifying Dieback in a Vulnerable Population of *Eucalyptus macrorhyncha* Using Remote Sensing

**Donna L. Fitzgerald** [1] , **Stefan Peters** [1] , **Gregory R. Guerin** [2] , **Andrew McGrath** [3] **and Gunnar Keppel** [1,4,*]

1   UniSA STEM, University of South Australia, Mawson Lakes, SA 5095, Australia;
    donna.fitzgerald@mymail.unisa.edu.au (D.L.F.); stefan.peters@unisa.edu.au (S.P.)
2   School of Biological Sciences, University of Adelaide, Adelaide, SA 5005, Australia;
    greg.guerin@adelaide.edu.au
3   Airborne Research Australia, Parafield, SA 5106, Australia; andrew.macgrath@airborneresearch.org.au
4   AMAP, Université de Montpellier, CIRAD, CNRS, INRAE, IRD, 34000 Montpellier, France
*   Correspondence: gunnar.keppel@unisa.edu.au; Tel.: +61-8-830-25137

**Abstract:** A disjunct population of red stringybark (*Eucalyptus macrorhyncha*) trees in South Australia is experiencing increasing amounts of dieback. Because the population is considered vulnerable to extinction, we investigated spatiotemporal vegetation changes, quantified the extent of dieback, and determined how topography influences dieback using aerial and satellite imagery. Classification of vegetation health status using hyperspectral aerial imagery indicated that 37% (accuracy = 0.87 Kappa) of the population was unhealthy and potentially experiencing dieback. When correlating this classification with a digital terrain model (DTM), the aspect and amount of solar radiation had the strongest relationship with the presence of unhealthy vegetation. PlanetScope satellite-derived, and spectral index-based analysis indicated that 7% of the red stringybark population experienced negative vegetation health changes during a five-year period (2017–2022), with positive vegetation health changes (9.5%) noted on pole-facing slopes. Therefore, our integrated remote sensing approach documented the extent and spatiotemporal dynamics of dieback, suggesting it could be applied in other studies. Topographical aspects exposed to high-solar radiation were particularly vulnerable to dieback, and pole-facing aspects demonstrated some recovery between droughts. The influence of topography and maps of vegetation health can be used to guide future management and restoration of the population.

**Keywords:** eucalyptus; dieback; remote sensing; stringybarks; satellite imagery; aerial imagery; conservation





## 1. Introduction

Drought-induced tree mortality has increasingly been reported in both the southern and northern hemispheres, with more extreme heat linked to global warming, which is believed to be exacerbating the effect of other environmental stressors [1–8]. Prolonged declines in annual rainfall [9] and an increase in the number of days with extreme temperatures appear to increase the risk of vegetation dieback [3,10], particularly in Mediterranean-type ecosystems [11], where rainfall is concentrated over autumn and winter with little to no rainfall during the hot summer periods [12,13]. Drought-driven dieback can be further compounded by anthropogenic alterations of the landscape [14], such as land clearance, changes in land use and general movement across the landscape by people and fauna [15,16]. Introduced pests and diseases [15,16] also impact how well the forest species can adapt to other environmental stressors.

Eucalypts are one of the dominant tree species in the Australian landscape, with over 800 species [17], providing the main canopy structure and habitat for many natural ecosystems. Eucalypts occur across an extensive range of aridity levels. Still, they are generally well adapted [1,8,18,19] to maintain water balance during periods of low soil moisture,

for example, displaying drought avoidance strategies, such as deep root systems [20], in some cases accessing groundwater [21], leaf shedding, and stomatal regulation [4]. However, some eucalypt forests do not show a high resilience to environmental pressures [22], with many eucalypt forests across Australia suffering the effects of dieback [2,3,23,24]. The known causes of dieback and forest decline are varied [2,25–27], including biological, drought, topographical location, and anthropogenic changes in the landscape.

Following the Millennium Drought (2001–2009) [28], increased levels of dieback, indicated by the reduced vigour of the tree's canopy, an increase in the number of dead branches held by the tree and, ultimately, the untimely death of the tree [29] were noted in a range of eucalypt species [2,3,22,25,30,31]. Given the importance of eucalypt forests for biodiversity and ecosystem services [32], the need to investigate the health and potential causes of dieback in South Australian eucalypt forests [24,33] has been identified. There is an urgent need to quantify the extent of dieback, monitor changes occurring within eucalypt forests over time and investigate the specific reasons for these changes to allow for possible countermeasures to be implemented in the future.

The red stringybark (*Eucalyptus macrorhyncha* F.Muell. ex Benth.) population in the Spring Gully Conservation Park (SGCP) north of Adelaide, South Australia, is at the most western extent of the species natural range. It has likely been disjunct and isolated from populations further eastward for tens of thousands of years [34]. It is considered a relic of a broader species distribution when the climate was more favourable [17]. After the Millennium Drought between 2001–2009 [28], an increased amount of dieback was observed within SGCP [31,35], and a successive drought, the Big Dry 2017–2019, resulted in further dieback [24,35], especially within the western area of SGCP [34]. Ground investigations of the red stringybark have indicated that monitored trees had a > 40% mortality rate from the beginning of monitoring in 2007 to the last year of monitoring, 2021 [34], with 20 % mortality after the Millennium Drought and 25% mortality after the Big Dry. With an increase in extreme weather events resulting from climate change [34,36,37], droughts are becoming hotter and more prolonged and have been termed climate change-type droughts [15,16]. Climate change impacts are forecast to be particularly severe in summer-dry Mediterranean-type ecosystems, placing this vulnerable red stringybark population at further risk of dieback [34,36].

Advances in remote sensing techniques using satellite or airborne imagery (derived using sensors on planes or remotely piloted aircraft (RPA) [38]) allow for the assessment of vegetation, including its health [39–43] and spatial-temporal dynamics [40,42,44] across the landscape. We here introduce an approach that integrates satellite and airborne imagery to investigate dieback in the South Australian red stringybark population. When completing this study, we aimed to (1) Determine if Remote Sensing techniques can complement previous ground investigations for conservation purposes, (2) Determine spatial and temporal patterns in vegetation health concerning droughts and topography, and (3) Determine the most suitable approaches of Remote Sensing to identify vegetation health changes with the view of broader application to other eucalypt forests. Our results indicate how maps on vegetation health, derived through remote sensing techniques, can provide important insights into the dieback process and can be used to guide future management and restoration of the population.

## 1.1. Understanding Dieback

Dieback, in a forest context, can be described as the decline of tree health with a loss in the vigour of crown cover and the premature death of trees following a stress event which can involve a single or combination of external and internal environmental factors [45], disease pathogens [26,46], insect attack [47] or changes in climatic conditions [2,22]. Climate change, particularly associated with increasing drought intensity, has been suggested to contribute to dieback [15,16,48–51]. For example, in Australia, where intense droughts occur regularly, climate change is creating an extreme set of drought conditions that lacks precedence in existing records [52]. Together with other environmental stressors,

such as the infestation of weakened trees by pathogens or insects, more intense droughts and extreme heat can create a positive feedback loop [53–55], ultimately resulting in the dieback of susceptible trees, such as the stringybarks [15,16,36,56]. Topography has been found to correlate with dieback occurrence after drought periods due to a reduction in water availability, especially within soils found on steep slopes and rocky outcrops [2]. When there is widespread dieback within a forest, changes in the species composition can occur [1,49,57,58], which can result in a more open ecosystem [59] susceptible to invasive weeds [60] and a loss of biodiversity for the area [61].

*1.2. Dieback Monitoring through Applied Remote Sensing*

Remote sensing allows non-invasive investigation across inaccessible [62] or fragile ecosystems, which is especially important for conservation efforts [63]. Passive remote sensing uses interactions with incident radiation (sunlight) and the target of interest (the earth's surface). It can take advantage of the differences in spectral bands in the visible (RGB: red, green, blue) versus the Near Infra-Red (NIR) part of the electromagnetic spectrum to quantify vegetation health [40]. When light hits vegetation, red and blue spectral bands are absorbed into the plant structure to a greater extent than green and much more than NIR. NIR bands, in particular, are reflected from chlorophyll [40]. The differences in reflectance values can be used to compute indices to indicate healthy versus stressed or dead vegetation [40]. Commonly used spectral measures to classify vegetation health status and monitor changes include the normalised difference vegetation index (NDVI), green NDVI (GNDVI), enhanced vegetation index (EVI) and soil-adjusted vegetation index (SAVI) [64,65]. The NDVI formula is based on healthy green leaves absorbing the red spectral band and unhealthy leaves scattering the NIR and green spectral bands [66]. The NDVI reduces the complexity of multiple bands in remotely sensed-raster imagery, letting comparisons of vegetation health occur [66,67].

High-resolution airborne hyperspectral imagery has emerged as a powerful tool in remote sensing [43,62,68,69], improving the identification of subtle variations in the continual spectral measurements of different vegetation [70,71], thereby allowing for the classification and quantification ($m^2$) of healthy vegetation versus stressed or dying vegetation [72]. Satellite imagery generally has a lower spatial resolution than airborne data and is subject to an inflexible observation schedule that takes no account of cloud or other time-related considerations. However, it provides near-global coverage with regular observations over a long period, providing a historical record of changes in the landscape [67]. The spatial resolution of freely available satellite imagery is typically > 2 m, whereas some commercial satellite products deliver < 50 cm resolution [65,68,69].

*1.3. Investigating Dieback Causes through Spatial Analysis*

Changes in topography have been found to impact vegetation health [73], with slope affecting soil moisture-holding capabilities and the aspect relating to the amount of solar radiation received [74–76]. Geoprocessing and the digital terrain model (DTM) can map the slope in degrees as a raster. The aspect can be derived by applying the compass direction of the downhill slope faces for each direction [77]. Further research has also shown the importance of solar radiation on vegetation health [75,76,78], in combination with other topography variables [79]. Solar radiation can be determined for a geographical area for a specific time by combining slope and aspect rasters with changes in the sun angle (daily and seasonal) to identify the shadows cast by the surrounding topography [77].

Effective management of the conservation efforts for the vulnerable red stringybarks within SGCP requires accurate knowledge of the extent and distribution of changes in vegetation health for the whole Conservation Park, along with an understanding of how local topography can influence the presence of unhealthy vegetation. In this study, we used remote sensing techniques to monitor forest health. Supervised classification of vegetation health will be applied to aerial hyperspectral imagery to identify the extent of dieback. A digital terrain model (DTM) was used to determine if topographic variables influence the

occurrence of dieback, and historical satellite imagery was compared to assess vegetation health changes over time.

## 2. Methods

### 2.1. Study Area

The study area is the SGCP, 5 km south of the Clare township and 136 km north of Adelaide, South Australia (Figure 1). SGCP was first established in 1961 to protect a remnant stand of red stringybarks (*Eucalyptus macrorhyncha*) in South Australia, where it is listed as vulnerable [33]. SGCP has a Mediterranean-type climate with a mean yearly precipitation of 540 mm, concentrated in winter and little rainfall over warmer summer months. It has a mean annual temperature of 21 °C with a mean summer (December–February) temperature of 28 °C based on input for Clare High School [80]. SGCP has an area of 4.11 km² with hilly terrain and steep-sided valleys [81] vegetated with an open woodland forest [17], which is dominated by the red stringybark. The red stringybark is an evergreen tree found in great numbers on the tablelands and western slopes of New South Wales and Victoria [20]; however, in South Australia, it is found only in the mid-north [82]. The red stringybark can be a single or multi-stemmed tree with fibrous brown bark; the lanceolate leaves are 150 mm long and 35 mm wide [17,82].

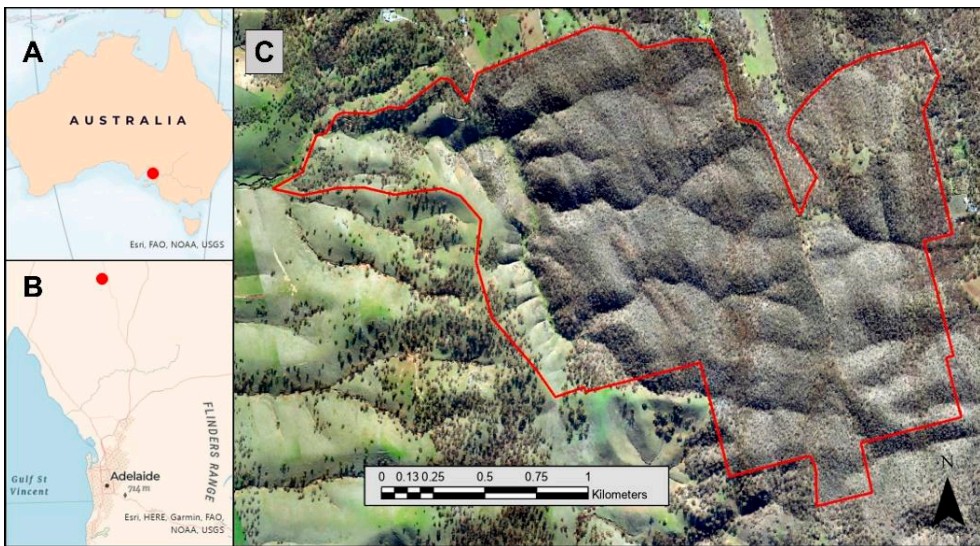

**Figure 1.** Location of the study area within Australia, (**A**) within South Australia, (**B**) and its surroundings. (**C**) Red outline indicating the boundary of Spring Gully Conservation Park.

### 2.2. Description of Imagery Data

One of Airborne Research Australia's HK-36 Eco-Dimona special mission aircrafts was used to capture airborne remote sensing data of the site. Specifically designed as a sensor platform, for this study, the aircraft carried a suite of remote sensing instrumentation, including a VNIR hyperspectral line-scanner, RGB camera, and a small-footprint, full-waveform lidar. Each instrument was coupled with an inertial measurement unit to locate the instrument in space and orientation for all data recorded, enabling the georectification of that data (Table S1). A flight in June 2022 covered the site with 25 parallel north–south flight lines separated by 100 m from a nominal 450 m above the terrain. This line spacing provides an overlap of image swaths from adjacent lines to ensure complete coverage with hyperspectral, RGB and lidar imagery.

Various satellite imagery was identified for the study area, including historical satellite imagery (Table S2). After preliminary analysis and initial processing of this imagery, we found that the earlier imagery from SPOT5 (2005, 2007) and QuickBird (2005) did not allow for a comparative analysis of differences in vegetation health. In particular, the 2007 SPOT5 and 2005 QuickBird imagery contained only RGB spectral bands, missing the NIR band,

and thus could not be used to create an NDVI raster. Furthermore, the 2005 SPOT5 satellite imagery had all required bands (RGB and NIR) but had a different spatial location and resolution than the more recent PlanetScope satellite imagery (2.5 versus 3 m resolution, respectively). This difference created a variation in the rasters, introduced errors and reduced the accuracy of the findings. The purchase of commercial satellite imagery with a 30 cm resolution was also investigated. However, it became cost prohibitive when several images from each time selected were required to cover the whole area of SGCP.

Therefore, considering the limitations of earlier satellite imagery and the need for freely available imagery from a single source, PlanetScope satellite imagery was chosen. It has a 3 m resolution, including RGB and NIR spectral bands, downloaded to include the SGCP area with a set criterion applied to reduce the effect of seasonal changes and visibility of the area. Only satellite imagery with 0% cloud cover was selected to ensure clear visibility. All imagery was collected in the month of March, i.e., after the hot, dry summer months (December to February), because at this time, the contrast between dry summer greases and the trees present would be more evident in the spectral responses of the vegetation.

### 2.3. Quantifying Vegetation Health in Space and Time

We used satellite and aerial imagery to assess vegetation health and explore dieback within the SGCP. This integrated remote sensing approach is illustrated in Figure 2 and was used to identify vegetation health status in 2022, determine the influence of topography on vegetation health status and identify vegetation health changes over time.

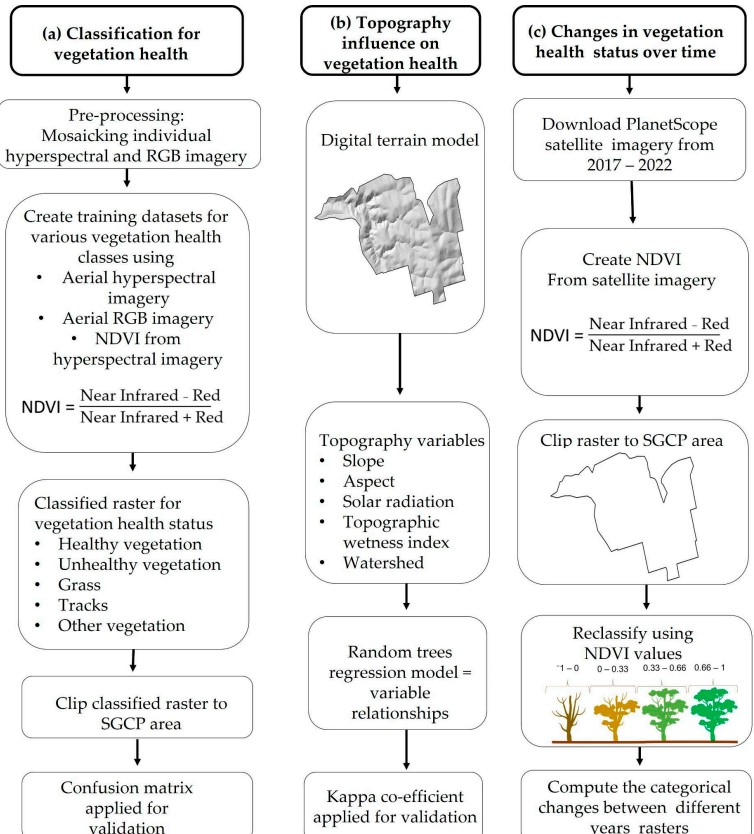

**Figure 2.** Illustration of the combined remote sensing approach applied in this study. (**a**) Supervised classification of very high-resolution hyperspectral aerial rasters to determine present (2022) spatial patterns of different stringybark health classes, (**b**) Integration of hyperspectral imagery and DTMs to determine how topography influences vegetation health. (**c**) Utilisation of PlanetScope satellite imagery time series to derive changes in vegetation health over time (2017–2022).

Individual hyperspectral rasters were mosaicked using Envi version 5.3 [71] to create one cohesive raster (Figure 2a) and to identify the vegetation health status in 2022. An NDVI raster created from the hyperspectral imagery identified differences in index values for vegetation health, with healthy vegetation depicted in white and areas of poor vegetation health defined as grey/black. Individual spectral band profiles of the identified vegetation were also used to determine vegetation health status as part of the classification process. The individual RGB frame images from the DSLR were mosaicked using AgiSoft Metashape version 1.5 [83] processing, creating a true-to-colour raster that could be used as another source for ground truthing of the results along with GNSS points gathered in the field earlier.

The hyperspectral, NDVI and RGB rasters were linked together using ENVI [71] to assist in identifying different areas of interest. To create a supervised classified raster identifying vegetation health in SGCP, the regions of interest (ROI) tool was used to locate areas of healthy vegetation, unhealthy vegetation, grass, tracks, and other vegetation areas within SGCP with a minimum of fifty samples each. For example, a healthy vegetation area would be seen as red for the hyperspectral imagery, white for the created NDVI raster and green for the RGB raster. In comparison, unhealthy vegetation in the hyperspectral rasters would decrease the vibrancy of the red color as grey to black for the created NDVI raster. Vegetation health status could also be identified on the RGB raster.

Using the Mahalanobis distance classification, a direction-sensitive distance classifier that assumes all class covariances are equal and are classified to the closest ROI [71], pixels within the hyperspectral raster could be allocated to the closest ROI class, which was then quantified numerically for comparative analysis. One hundred [84] random accuracy points were created in ArcGIS Pro [77] to verify the accuracy of the modelling classification. A Kappa co-efficient [85] was applied to assess the results' validity by creating a Confusion Matrix in ArcGIS Pro [77].

### 2.3.1. Topography Influences in Vegetation Health Status

To depict the underlying topography of SGCP, a digital terrain model (DTM) was used to derive topography variables, such as slope in degrees, aspect, solar radiation ($WH/m^2$) and topographic wetness index (TWI), which were computed using ArcGIS Pro, [77] and a watershed raster created in Global Mapper [86]. The slope analysis determines the steepness in degrees of the SGCP area, measurements ranging from 0 to 90 degrees, with a flat surface representing zero and a higher value representing a sloping surface with a greater degree angle. Aspects were obtained from the inclining facing slope and defined by differences in numerical numbers (0–360) for each aspect, depicted in contrasting colours. Solar radiation levels (watt hours per square meter ($WH/m^2$)) affecting SGCP were identified using a sky resolution of 200 m over 160 days from January 2022 to April 2022. The raster indicates the different levels of solar radiation occurring within SGCP with attribute values from 44,139 $WH/m^2$ to 667,560 $WH/m^2$.

The topographic wetness index (TWI) dictates residual moisture levels for the SGCP area [87] following several steps utilising geo-processing tools in ArcGIS Pro [77]. Starting with the DTM, the hydrology tool 'fill' was used to determine the flow direction, followed by the flow accumulation tool. The surface slope was determined with an output measurement in degree, which allowed for the calculation of Radians of slope ((slope in degree*×1.570796)/90) and Tan slope (con(slope > 0, tan(slope), 0.001). Flow accumulation was then scaled ((flow accumulation + 1) *cell size). Finally, a TWI (flow accumulation scaled / tan slope) could be created to indicate residual moisture levels of SGCP. A watershed raster was created to identify the drainage networks within SGCP using the watershed geoprocessing tool in the Global Mapper program [86].

The classified raster, indicating unhealthy vegetation created in ENVI [71], was overlaid onto each topographic variable map to identify correlations visually. A supervised machine learning process in geoprocessing was used to understand the relationship between vegetation health (healthy/unhealthy) and the variables (slope, aspect, solar radiation,

TWI, and watershed). A random forest regression model [88], used in ArcGIS Pro, created 100 decision trees with 100,000 samples. The resulting sum for all the variables' values equals 1, with a large number indicating that the variable is more correlated to vegetation health [89].

2.3.2. Vegetation Changes over Time Using Satellite Imagery

The NDVI formula using spectral band 4 (NIR) and band 1 (RED) was deemed to be the most appropriate vegetation monitoring formula to use, as confirmed by previous studies [66,67,90,91]. A series of geoprocessing steps (Figure 2c) to identify if vegetation changes in health for SGCP were implemented using the ArcGIS Pro [77] and PlanetScope satellite rasters for each year (2017–2022), which were clipped to the SGCP boundary.

One method used was to analyse the difference in the number of pixels whose NDVI moved in a negative or positive direction from 2017 to 2022, allowing for the interpretation of change on a continuous scale through a histogram. Applying a threshold of 1 standard deviation NDVI change value could imply a significant shift in the vegetation health status. A limitation of this method is that trivial factors can influence NDVI pixel variations through time, such as changes in plant arrangement and weather conditions [66], and that a pixel cell represents spectral values in a grid in which the boundaries lack real-world correspondence [92]. Therefore, to identify ecologically significant thresholds of change to vegetation health (i.e., unhealthy, moderate, healthy), the NDVI rasters derived from each year were reclassified into four hard vegetation health levels with assigned values: dead (−1–0), unhealthy (0–0.33), moderate (0.33–0.66) and healthy (0.66–1).

Using the compute change raster tool in the geoprocessing area of ArcGIS [77], selected rasters from each year were compared against each other to determine the level of change in the overall vegetation health of the SGCP. Positive vegetation health changes within SGCP were identified as moving from unhealthy–moderate, unhealthy–healthy and moderate–healthy. Negative vegetation health changes within SGCP were identified as moving from the categories: moderate–unhealthy, healthy–unhealthy and healthy–moderate. When there was no movement from one category to another, we noted that no change had occurred. Geoprocessing steps created an attribute table, indicating how much area ($m^2$) was affected by vegetation change between the periods chosen. Validation tools were then applied to results as a form of quality control [89] using geoprocessing tools to quantify the accuracy of remote sensing techniques.

The ground-truthing handheld GPS points gathered in the field [34] for validating the results for changes over time using PlanetScope satellite imagery were found to be unreliable. The variances in the accuracy of the GPS location results could be out by up to 5 m depending on how many satellites were present, with steep areas showing a further decrease in accuracy. Discrepancies occurred when combined with the spatial resolution of the satellite image at 3 m, resulting in difficulty in classifying vegetation. Another problem identified was that GPS points were only taken along the previously monitored transect lines and not throughout the SGCP, reducing the ability to validate all regions of the Conservation Park. Therefore, it was decided that ground-truthing of all images would be completed with a high-resolution 50 cm RGB imagery, which has become a widely accepted method in remote sensing [93–95].

## 3. Results

### 3.1. Vegetation Health Status in 2022

The classification of vegetation (Figure 3) within SGCP suggested that 47% (1.9 km$^2$) is healthy vegetation, 37% (1.5 km$^2$) is unhealthy vegetation, 8% is grass, 7% is other vegetation, and 1% is tracks. The confusion matrix suggested accurate classifications (Kappa = 0.87) with the accuracy of identifying unhealthy vegetation correctly being 0.90.

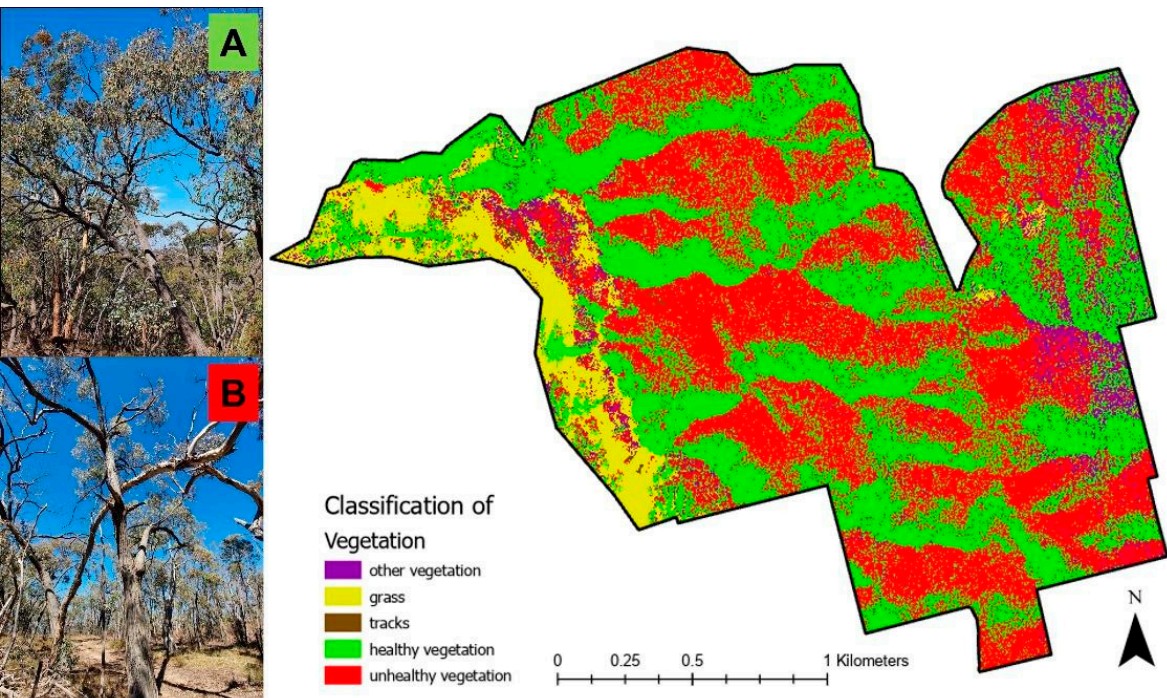

**Figure 3.** Classified vegetation health status map using 2022 aerial hyperspectral imagery at 50 cm resolution of the SGCP. Images (**A**,**B**) provide an indication of how the healthy (47%) and unhealthy (37%) vegetation classes would look on the ground, respectively.

### 3.2. Topography Influences on Vegetation Health Status

The terrain is very hilly and steep in some parts of the SGCP and even in others (Figure 4a). The slope degrees range from zero, indicating flat areas mainly located at the top of the elevated regions, to very steep areas of 84 degrees (Figure 4a) north and south facing. The prevailing aspects in SGCP are orientated north and south (Figure 4b). SGCP is exposed to high-solar radiation, especially along the northwest-facing slopes (Figure 4c). These slopes are exposed to the most increased solar radiation and have high levels of unhealthy vegetation.

The Topographic Wetness Index (TWI) and watershed rasters indicate that SGCP has no signs of large streams or accumulated catchment areas. The drainage network consists of many small tributaries that dissipate into the surrounding area outside the SGCP boundary. There is not a direct relationship between TWI and watershed with the presence of unhealthy vegetation.

The random regression model (Figure 4e) found that solar radiation had the highest ($R^2 = 0.37$) correlation coefficient with vegetation health, followed by aspect ($R^2 = 0.31$) and slope ($R^2 = 0.20$). TWI ($R^2 = 0.03$), whereas watershed ($R^2 = 0.09$) showed no correlation. Overall, including all five variables, the random trees regression model suggested that topographic variables and their derivatives explain a considerable variation in the observed vegetation health status ($R^2 = 0.71$).

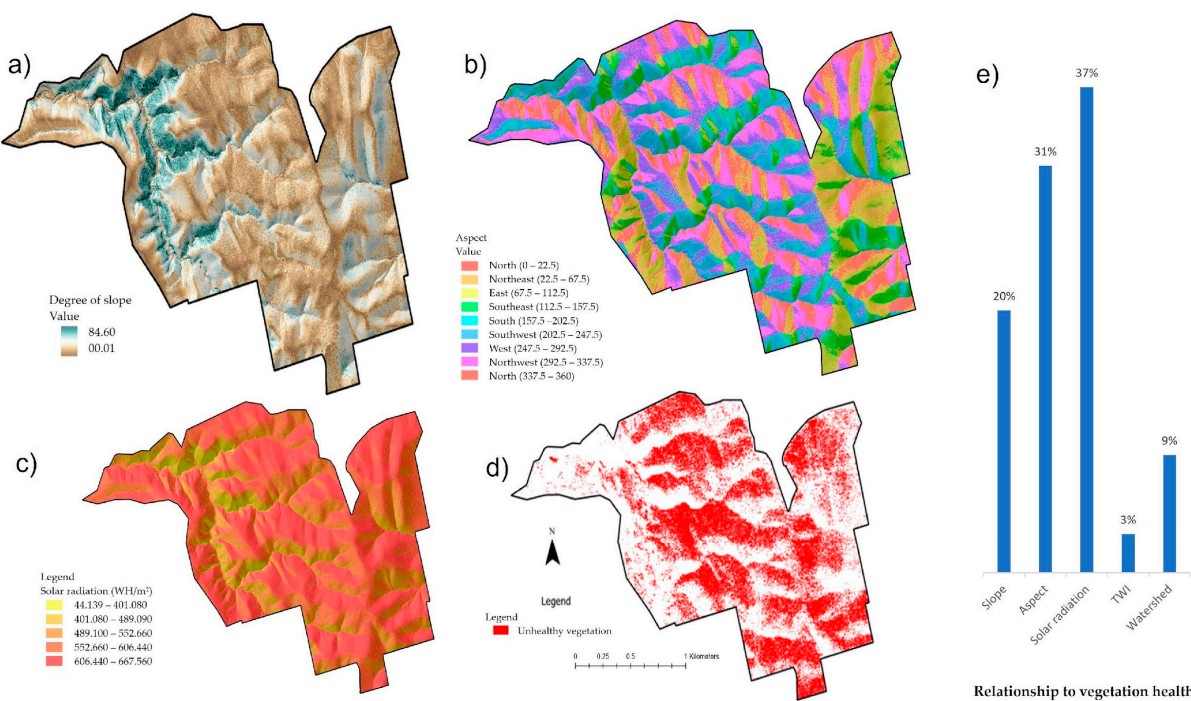

**Figure 4.** Digital terrain model (DTM) created June 2022 (**a**) Computed degree of the slope with the green area indicating a greater degree of slope. (**b**) Computed aspect image highlighting northwest-facing slopes in pink. (**c**) A computed solar radiation image indicates red areas exposed to high solar radiation levels from January to April 2022. (**d**) Unhealthy vegetation classified image showing the relationship with topography variables. (**e**) The relationship of variables with vegetation health was identified through a random regression model in ArcGIS Pro, indicating that solar radiation at 37% and aspect at 31% had the highest correlation with vegetation health.

### 3.3. Vegetation Changes over Time Using Satellite Imagery

To assess overall changes in vegetation health between 2017 and 2022, the NDVI (derived from PlanetScope satellite imagery) and associated histograms (Supplementary Figure S1) were compared for differences in vegetation health. A total of 49% of the pixels moved in a negative direction (suggesting a decrease in health) from 2017–2022, and 51% moved in a positive direction (implying an increase in health). As many trivial factors can influence pixel variations, including plant arrangement and weather conditions, a threshold was applied using the standard deviation of 0.09 from the histogram. This resulted in 16.8% of pixels moving into a negative value and 14.6% into a positive value (Figure S2).

Overall, positive changes in vegetation health categories (see Table S3) occurred in a positive direction for 9.5% (0.39 km$^2$) of the area, mainly on the south-to-southwest downward-facing slopes. The percentage of priori health categories that shifted in a negative direction was 7% (0.29 km$^2$), occurring across the whole SGCP, not just isolated to certain areas or small pockets (Figure 5). The main regions of negative category change occurred along the east-to-northeast downward-facing slopes. The remaining 83% (3.43 km$^2$) of SGCP stayed within the same health category from 2017–2022, although it is likely that change within categories also occurred.

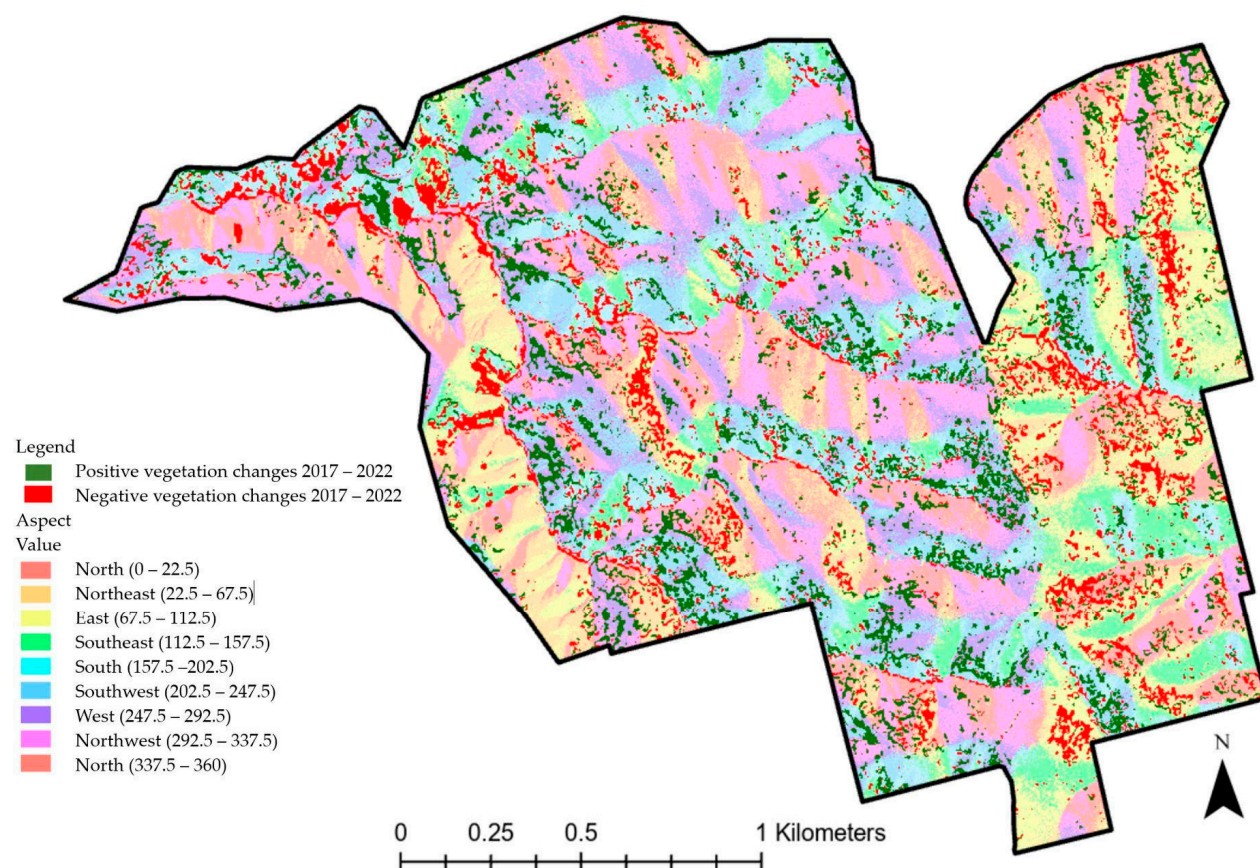

**Figure 5.** SGCP vegetation health changes were identified using PlanetScope satellite imagery from 2017–2022. We found that 9.5% of SGCP shifted vegetation health categories in a positive direction, shown in green, and 7% shifted in a negative direction, shown in red. For the remaining 83% of SGCP, vegetation health categories remained within the original vegetation health categories but may have changed within those categories.

A decrease in rainfall from 2017–2019—the Big Dry period (Figure 6)—was associated with continual negative vegetation health change. When there was an increase in summer and autumn rainfall in 2020, there was an increase in positive vegetation health changes between 2020–2021. These improvements in vegetation health were mainly located in areas of south-facing downward slopes (Figure 7). However, after a reduction in summer rainfall amount during 2021, positive vegetation health changes within SGCP decreased between 2021–2022.

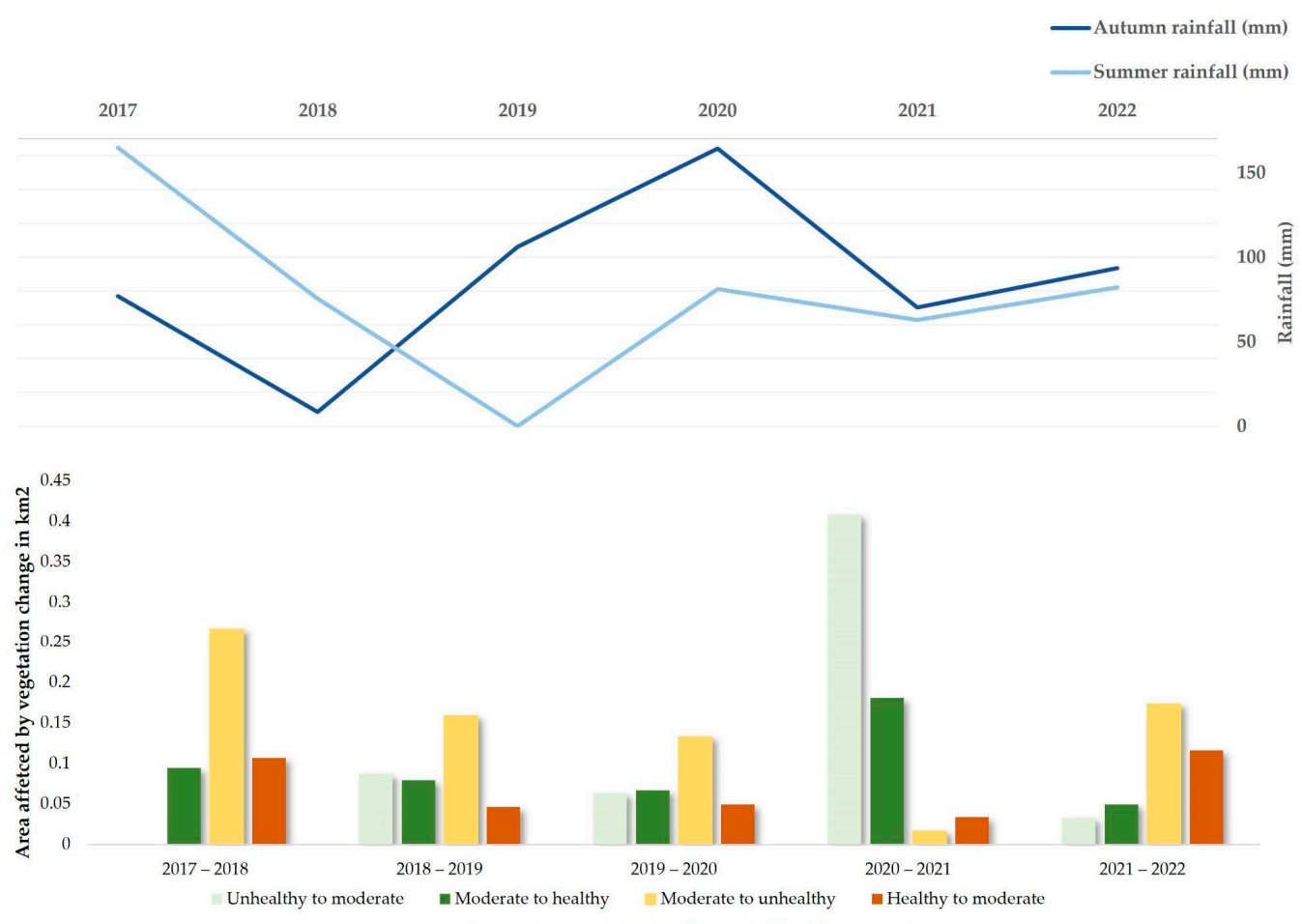

**Figure 6.** Changes in vegetation health in relation to seasonal rainfall. Changes from one vegetation health category to another, area of SGCP affected in km$^2$ between 2017–2018, 2018–2019, 2019–2020, 2020–2021 and 2021–2022 using PlanetScope satellite imagery. Rainfall (mm) amount occurring from December–February (summer) and March–May (autumn) for each year from 2017–2022. Note that the amount of rainfall is indicated between the interannual changes in vegetation health, i.e., the rainfall amounts for 2018 are shown between the 2017–2018 and 2018–2019 vegetation changes.

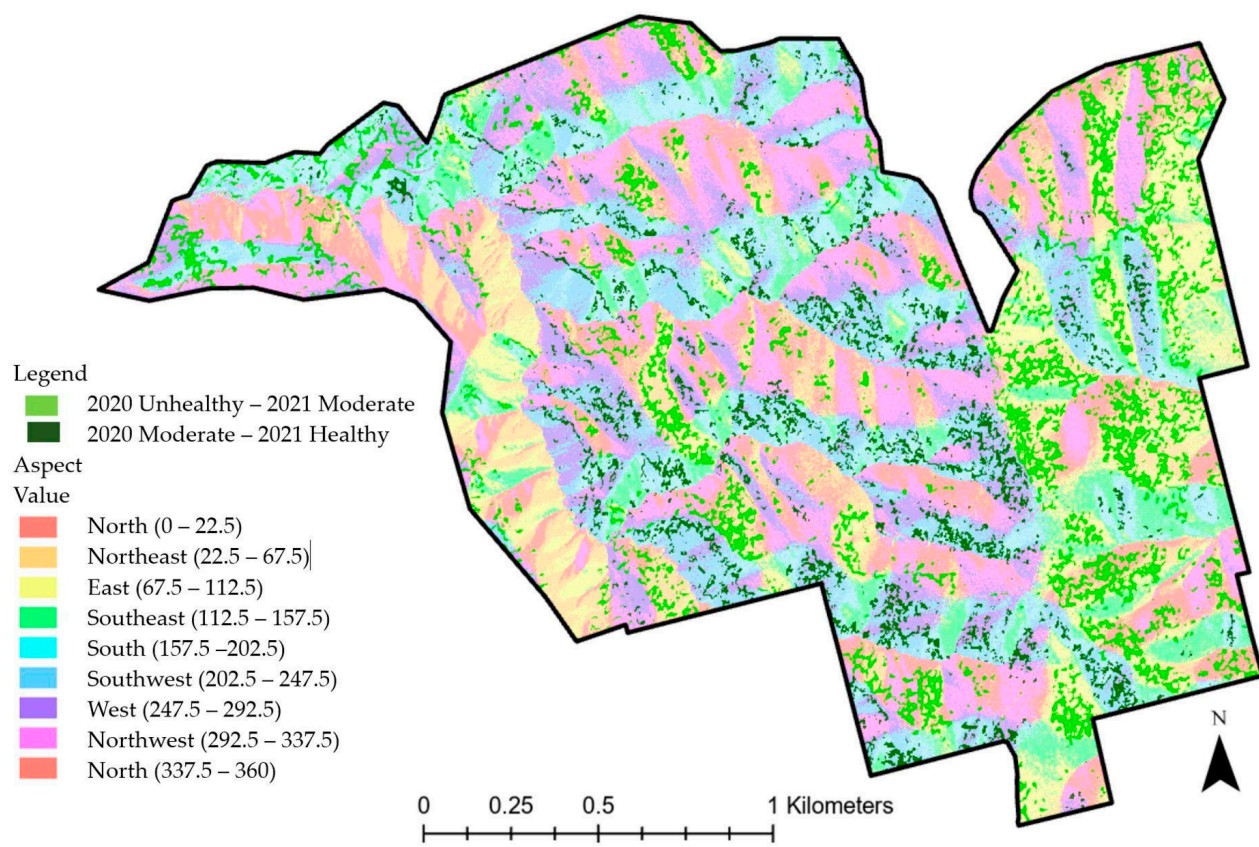

**Figure 7.** Positive vegetation health changes after high 2020 summer and autumn rainfall have occurred mainly on the south-facing slopes.

## 4. Discussion

Our findings illustrate that an integrated remote sensing approach, using both aerial and satellite imagery, can provide detailed spatial and temporal insights into the process of vegetation dieback. This research contributes to the knowledge of dieback of the red stringybarks in SGCP using a combination of remote sensing methods to identify changes in vegetation health over time, identifying the extent of changes and determining topographical influences in the occurrence of dieback. This research builds on and confirms results from previous monitoring of the occurrence of dieback within the SGCP, which showed that the vulnerable population lost > 40% of its individuals and biomass through droughts since 2005 using on-the-site field survey methods [34,35]. When using hyperspectral imagery to classify vegetation health in 2022, 37% of SGCP was considered unhealthy, suffering dieback effects. Therefore, both studies suggest that the red stringybark ecosystem in SGCP is in decline, with extensive dieback occurring over the last two decades.

Using hyperspectral imagery and the DTM, we illustrated dieback's spatial and temporal dynamics in a population of vulnerable eucalypt species. Topography is known to affect the health of vegetation in the landscape [34,96]. On this basis, we expected that steeper slopes would correlate with poorer vegetation health. Based on the assumption that a higher rainfall run-off occurs on steeper slopes [2,76], the higher gradient reduces the time for water absorption into the soil [79]. However, our results show that unhealthy vegetation is not limited to steep areas in the SGCP. Aspects were found to be related to unhealthy vegetation within the SGCP. Our results confirmed an increase in the dieback of red stringybark trees on the northern and western slopes [34,35]. Northwest-facing slopes in the SGCP are exposed to high-solar radiation levels between January and April and have a higher level of unhealthy vegetation than south-facing slopes. Aspects and solar radiation had a strong relationship with vegetation health in the random trees regression model, which has also been observed in other systems [79].

Hence, topography significantly influences vegetation health in SGCP, and such information is highly relevant for restoration planning, highlighting the potential of remote sensing approaches to assist restoration efforts. In particular, healthy vegetation was predominantly found on south-facing slopes; therefore, such sites could be more suited to future revegetation programs. The long-term success of sampling plantings is more likely when tube stock is planted on the south-facing slopes exposed to lower solar radiation than on the north-facing slopes.

Previous studies have identified that water and its flow impact vegetation [78,79]. The identified low residual moisture levels of the SGCP were found using the TWI. The watershed image showed little correlation to vegetation health status, even though drought (below average rainfall) has been implicated in dieback within SGCP [36]. It should also be noted that the elevated position of SGCP, compared to the surrounding landscape, reduces the water-holding capacity of the area due to gravitational drainage. Further investigation using field methods of this topography variable may provide more significant insights.

Using PlanetScope-derived satellite imagery, we showed temporal changes to the vegetation health status within SGCP from 2017–2022 using shifts in pixel NDVI > 1 standard deviation and shifts between hard threshold vegetation health categories. Comparing the NDVI rasters with both assessment methods indicated a decline and recovery in the red stringybark population from 2017–2022. A limitation in creating hard boundaries for vegetation health status was that it might not reflect physiological thresholds, and therefore may introduce bias because a shift in NDVI may or may not register as a change just because of the starting position [92]. This could potentially be resolved with further field accuracy assessment of vegetation health thresholds and their corresponding NDVI values [97].

A further limitation of using satellite imagery with the spatial resolution of 3 m was the inability to accurately quantify the extent of the vegetation status of SGCP, due to the inability to identify individual healthy versus unhealthy trees. Commercially available high-resolution imagery could potentially resolve this issue [98]. However, costs were prohibitive. To complement the temporal satellite-derived data, aerial imagery was used to improve the spatial resolution for vegetation monitoring in SGCP to 50 cm. This imagery allowed for identifying subtle variations in the vegetation's continual spectral measurements [63,64]. The results clearly show that the health and extent of the red stringybark ecosystem are in decline, with 37% of the SGCP considered unhealthy, which confirms field results from previous studies [34,35].

Remote sensing has emerged as a powerful tool for portraying landscape changes and can be quickly understood by many stakeholders, including those without a background in research [63]. Creating explicit imagery in the form of maps representing the monitored landscape [40], identifies vegetation health status, along with an attribute table that can be used for comparative statistical analysis. When applying an integrated remote sensing approach to determine vegetation health status, the combined results allow for a broader understanding of where and when the dieback occurs while identifying areas for the red stringybark's regenerative conservation efforts. Our results highlight areas of vegetation decline within SGCP since 2017 and recovery of vegetation health on pole-facing slopes. Though not supported by direct physiological measurements, it appears to be predominantly driven by drought and heat, given the correlation of dieback with topographic position and drought events [34,35]. Our results indicate that the dieback process in SGCP is complex in space and time, with areas experiencing high-solar radiation, mostly having poor vegetation health and showing no recovery after high summer and autumn rainfall in 2020, while pole-facing slopes displayed better health and improved vegetation health after rain.

Dieback in other eucalypt species has been noted after prolonged and acute drought periods [2,3,26]. However, the ultimate death of trees due to hydraulic failure is not a linear process [99], but rather one with critical thresholds of water potential at which catastrophic xylem embolism occurs [100]. Eucalypts can produce epicormic-bud growth [101] after stress events such as drought [22], which can explain recovery after increased rainfall

improves crown cover, which may be reflected in the NDVI values. In SGCP, intermittent recovery of the red stringybark occurred when there was an improvement in vegetation health, as noted in the NDVI after high rainfall in 2020. Unfortunately, with a decrease in rainfall the following year, vegetation improvement (new crown growth) could not be sustained when looking at the NDVI values. The forecast rise in drought intensity and frequency due to climate change [24,60,102,103] does not bode well for the red stringybark. Therefore, large extents of this unique population are at high risk of further negative vegetation changes over time.

Another limitation of this research was the resolution of freely available PlanetScope imagery of 3 m, which generally did not allow for identifying individual trees and did not cover the period before and during the Millennium Drought (2001–2009), when considerable dieback occurred [34,35]. Therefore, the extent of mortality caused by the Millennium Drought cannot be estimated from the PlanetScope imagery available. Results on vegetation changes between 2017–2022, therefore, are not representative of the full decline experienced by this population. Commercially available historical satellite imagery with higher resolution than the PlanetScope imagery, such as the imagery from Worldview at 30 cm resolution, could provide insights into the impacts of this earlier drought.

Using complementary remote sensing techniques and adapting the methods employed here, spatiotemporal patterns in dieback can be assessed for other eucalypt species, such as two other stringybark species, *E. baxteri* and *E. obliqua*, that are also experiencing dieback [36,104]; these are found in the Mount Lofty Ranges. Findings will help guide environmental policy decisions for the future of these stringybark species. Future studies involving SGCP should include repeat gathering of hyperspectral imagery, providing greater accuracy in comparing changes and the extent of vegetation health changes over time. The available DTM could assist in predicting future locations of unhealthy red stringybarks by adapting the developed methods and understanding topographic variables and their relationship to vegetation health.

## 5. Conclusions

By combining satellite and aerial imagery, we have shown spatial and temporal changes in vegetation health within SGCP, creating a clearer picture of the red stringybark dieback. This integrated approach identified the extent and distribution of red stringybark dieback, highlighting aspects exposed to high-solar radiation as areas of severe decline. Positive changes in vegetation were found in pole-facing aspects, areas subjected to lower solar radiation levels. This research complements results from previous fieldwork studies, reinforcing that the red stringybark population in SGCP remains particularly vulnerable to increasing droughts. Maps of vegetation health can guide future management and restoration of the population, identifying areas for red stringybark tree restoration. Planting seedlings on south-facing slopes, which experience a reduced amount of solar radiation, may help the survival of this unique red stringybark population. Therefore, the methodology employed here has great potential to be adapted to monitor the vegetation health status of other eucalypt species.

**Supplementary Materials:** The following supporting information can be downloaded at: https://www.mdpi.com/article/10.3390/land12071271/s1, Table S1: Airborne Research Australia (ARA) instruments used for aerial monitoring; Table S2: Range of satellites (commercial and freely available) for gathering multispectral satellite imagery; Table S3: Changes within vegetation health thresholds; Figure S1: Histograms for (a) 2017 and (b) 2022 NDVI created from PlanetScope satellite imagery. (c) Histogram representing the difference between the 2017 and 2022 NDVI, with results showing 49% of pixels moving negatively and 51% moving positively; Figure S2: SGCP vegetation health changes were identified using PlanetScope satellite imagery from 2017–2022. (a) Using the standard deviation threshold of the difference in NDVI histogram to understand changes in pixel values for vegetation health. (b) For 68.60% of SGCP, NDVI shifted within 1 standard deviation, interpreted here as trivial variation. (c) Significant shifts in vegetation health, interpreted here as greater than

1 standard deviation, occurred for 14.60% (positive; shown in green) and 16.80% (negative; shown in red) of pixels.

**Author Contributions:** Conceptualization, D.L.F., G.K. and S.P.; methodology, D.L.F., S.P. and A.M.; software, D.L.F., S.P. and A.M.; validation, D.L.F., G.K. and S.P.; formal analysis, D.L.F.; investigation, D.L.F. and A.M.; resources, D.L.F.; data curation, D.L.F. and A.M.; writing—original draft preparation, D.L.F.; writing—review and editing, D.L.F., G.K., S.P., G.R.G. and A.M.; visualisation, D.L.F.; supervision, G.K., S.P. and A.M.; project administration, D.L.F., G.K., S.P. and G.R.G.; funding acquisition, D.L.F. and G.K. All authors have read and agreed to the published version of the manuscript.

**Funding:** This research was partially funded by Airborne Research Australia, The Playford Memorial Trust scholarship, a Nature Foundation Grant and a NRM Research and Innovation Honours Scholarship. G.K. was partially funded and supported by the Montpellier Advanced Knowledge Institute on Transitions (MAK'IT) as part of a Visiting Scientist Fellowship.

**Data Availability Statement:** The data presented in this study are available on request from the corresponding author. The data are not publicly available due to privacy restrictions.

**Acknowledgments:** We acknowledge the traditional owners of Clare Valley, the Kaurna and Ngadjuri people. We would also like to acknowledge the valuable support of Airborne Research Australia, which collected high-resolution aerial imagery at a minimum cost.

**Conflicts of Interest:** The authors declare no conflict of interest. The funders had no role in the study's design; in the collection, analyses, or interpretation of data; in the writing of the manuscript; or in the decision to publish the results.

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
