# Peer review of "Quantifying Dieback in a Vulnerable Population of Eucalyptus macrorhyncha Using Remote Sensing"

_land, doi:10.3390/land12071271_

Round 1

Reviewer 1 Report

see attached

Author Response

This is an exciting piece of work and I am not aware of other papers reporting on the remote sensing of Drought-induced tree mortality (DITM) in eucalypt forest. Therefore the topic is worthy.

Author responseThank you.

My problem with this study is with its framing, and the same goes for Keppel et al., a foundation paper on the same phenomenon. The authors are proposing that the tree mortality event for red stringybark at SGCP is a climate-change event and might cause the extinction of this population if it becomes exaggerated in the future.

Author responseWe understand and agree with the reviewer’s assessment. This research focuses on the effect of drought on the SGCP red stringybarks and does not investigate the impact of climate change. We have made several changes to the text that hopefully highlight that we are looking at the impact of drought on vegetation, but that climate change has been shown to increase the intensity of drought conditions - please see alterations in the text on pages 1 – 4. However, we would like to highlight (and do this a bit more in the manuscript now) that the SGCP population occurs in a Mediterranean-type climate, with most of its rainfall occurring in the winter and little rain over the summer months. Therefore, while overall mean rainfall for the year may not indicate a rainfall deficit, the timing of this rainfall is important. In a Mediterranean climate, if most of the annual rainfall occurs within the cooler months and there is an extended period of little or no rain during the summertime, the vegetation population will be placed at further risk if there is an increase in droughts, as predicted by climate change. Potentially this will have a significant impact on this population. Projected climate change with increasing drought and above-average temperatures at sustained levels, therefore, could have a negative effect on this vulnerable population, located in the most western extent of its natural range, with its closest population occurring in Dubbo, NSW.

I think it is remiss to arrive at this conclusion without even examining the rainfall record. I stripped this from the SILO site (attached) and a three-year index reveals an equivalent three-year rainfall deficit around 1930.

Author responseThank you for pointing this out and sharing the rainfall graph obtained by SILO data. As mentioned above, this article focuses on investigating dieback in relation to drought rather than determining if climate change contributes to the observed decline. We, therefore, decided that further extending the climatic history of the study site was of limited relevance for this paper (and did not follow up on this aspect).

This suggests that the event could be a natural phenomenon in keeping with the careful analysis of these events by Fensham and co-workers. The following papers and other referenced within are relevant. These papers provide a framework for assessing the recovery potential of species affected by DITM. There is also a framework in these studies to examine DITM in relation to species ranges, and therefore whether they are critical for determining distributions.

The graph provides other points of interest worthy of consideration.

I would suggest that the authors take the opportunity here to reconsider their understanding of DITM, and present a more nuanced interpretation.

Author response: Thank you for sharing with us the other papers on our topic, which have been reviewed and added to the referencing for this paper. In this novel study of dieback within a eucalypt forest, our focus was to use remote sensing techniques to assist with creating frameworks for conservation areas. For future studies, other frameworks, as suggested, may also be incorporated for a broader view. However, it was beyond the scope of this study to do so.

Australia is a land of drought and flooding rain, and the fact that these events are ‘normal’ should be the null hypothesis, before evoking climate change. Although it is a bit ambiguous it seems that there is more of this stand showing positive health compared to negative health even during drought.

How do the authors conclude that this is an ecosystem in decline?

What about recovery?

Author responseAs mentioned above, we have modified the text to indicate that droughts are common in the Australian landscape but have also highlighted the well-documented fact that drought conditions have become more intense than they have been for centuries (Allen KJ, Verdon-Kidd DC, Sippo JZ, Baker PJ. 2021. Compound climate extremes driving recent sub-continental tree mortality in northern Australia have no precedent in recent centuries. Scientific Reports 11: 18337). As mentioned earlier, this paper focuses not on whether climate change contributed to the observed dieback but on remote sensing approaches to better understand dieback as a result of drought. While we agree that such events being normal should be the null hypothesis, we believe that this is a non-issue in our (now hopefully much clearer) storyline.

Regarding the issue of population decline, we respectfully disagree. Our high-resolution 2022 aerial hyperspectral imagery at 50 cm resolution indicates that 37% of the SGCP supports unhealthy vegetation, which corresponds well with >40% mortality in field-measured mortality of tagged trees since 2005.

While the population likely is in a long-term decline, we agree that the results presented for the 2017-2022 satellite imagery do not show a decline in population health. We, therefore, now do not refer to the satellite imagery time series showing population decline anymore but maintain that the population is declining based on the large percentage of unhealthy vegetation and the field surveys published earlier.

If ecologists fail to establish the background to this phenomenon then we will be flat-footed when climate change kicks in! There is a highly relevant literature on drought-induced tree mortality in eucalypt forest and woodland in Australia that has been overlooked. This research suggests these events are natural disturbance phenomena that are not necessarily a symptom of anthropogenic climate change.

Author responseThank you for sharing with us the other papers on our topic, which have been reviewed and added to the referencing for this paper.

Reviewer 2 Report

The work is relevant and well-written. The authors should improve the quality of Figure 6

Author Response

The work is relevant and well-written.

Author response: Thank you.

The authors should improve the quality of Figure 6

Author response: Thank you for pointing this out. The reviewer is correct, and we have improved the layout of Figure 6 (page 12 in the manuscript) to improve the clarity of the information portrayed.

Reviewer 3 Report

Dear Authors,

the article entitled: Quantifying dieback in a vulnerable population of Eucalyptus macrorhyncha using remote sensing. presents a very important issue. The article is very interesting and precisely described

However, in the publication make the following changes:

In the introduction, briefly characterize the individual chapters.

The novelty of the study is missing from the paper. What are the innovations brought in by this study? This must be added to the introduction section?

To sum up, after taking into account the above amendments (minor revision), I suppose that this article is suitable for publication in Land.

Author Response

The article entitled: Quantifying dieback in a vulnerable population of Eucalyptus macrorhyncha using remote sensing. presents a very important issue. The article is very interesting and precisely described.

Author response: Thank you.

In the introduction, briefly characterize the individual chapters.

Author response: As suggested by the reviewer, we have outlined the chapters in the introduction (pages 2-3 in the manuscript).

The novelty of the study is missing from the paper. What are the innovations brought in by this study? This must be added to the introduction section.

Author response: As suggested by the reviewer, we have noted the importance of this study in the introduction (pages 2-3 in the manuscript). This study introduces an approach incorporating two remote sensing techniques to understand dieback in a eucalypt forest for analysing spatial and temporal vegetation changes. It also shows that our remote sensing approach can provide important new insights into the process of dieback and can be used to guide future management and restoration of the population.